# A Novel Approach Proposal for Estimation of Ultimate Pile Bearing Capacity Based on Pile Loading Test Data

İsa Vural [1], Halil Kabaca [2] and Semiha Poyraz [3,4,*]

1   Faculty of Technology Civil Engineering, Sakarya University of Applied Science, Sakarya 54100, Turkey; ivural@subu.edu.tr
2   Uzo Sadikoglu Construction Limited Company, Bursa 16245, Turkey; halilkabaca@gmail.com
3   Graduate Education Institute, Sakarya University of Applied Science, Sakarya 54100, Turkey
4   Vocational School, Bilecik Seyh Edebali University, Bilecik 11100, Turkey
*   Correspondence: semiha.poyraz@bilecik.edu.tr; Tel.: +90-2282141336

**Abstract:** Determining the ultimate bearing capacity of piled foundations has been one of the most important problems in geotechnical engineering. It has been observed that the pile loading test evaluation methods based on the mathematical model give values far away from the failure load, even in piles that have reached the failure state. For this reason, it is aimed at developing a new mathematical pile loading test evaluation method based on the load-settlement curve. It has been thought that the main problem with the pile loading test evaluation methods based on other mathematical models giving values far from the failure load is that the methods iterate excessively. The new method proposed in the study was developed by taking this situation into account. The performance of the proposed new method was investigated using eight pile loading tests conducted in various provinces of Turkey. In order to verify the reliability of the newly developed method, the study was completed by applying multiple comparison tests with other methods in the literature (theoretical methods, finite element analysis methods, and pile loading test evaluation methods). According to the applied analysis of variance, it was concluded that the proposed new method remained within the 95% confidence interval and was usable.

**Keywords:** piles; pile bearing capacity; pile loading test; bored pile; foundation

## 1. Introduction

The safe transfer of loads from structures to the ground is very important in terms of geotechnical engineering. If the foundation cannot bear these loads, serious damage may occur to the structures and may even result in the loss of life. Pile foundations, which are the deep foundation type, are a solution that can be applied in cases where the shallow foundations are insufficient in terms of bearing capacity due to heavy structural loads and insufficient ground conditions. There are many methods developed for the efficiency of piles. The load transfer mechanisms of pile foundations need to be well known because the load transfer mechanisms of piles are extremely complex [1]. In general terms, the vertical bearing capacity of a single pile can be calculated by testing the soil that the pile is in with soil tests or by determining its mechanical properties by experimental methods [2].

The Standard Penetration Test (SPT), which is one of the oldest soil tests and is still widely used today, is a test that determines soil density and consistency as well as helps in the calculation of pile bearing capacity. There are commonly used methods based on SPT data, such as Meyerhof, Bazaraa and Kurkur, Decourt [3–5]. Meyerhof is a method based on empirical coefficients and SPT data for determining the pile bearing capacity in coarse-grained soils, developed taking into account the presence of driven or drilled piles [3]. There are empirical coefficients based on SPT data using the Bazaraa and Kurkur method, which was developed by studies on Egyptian soils [4]. In the Decourt method, there are empirical coefficients that can be used in driven and bored piles developed for

both fine- and coarse-grained soils depending on SPT values [5]. Studies on 81 cases in Vietnam were conducted by Huynh et al. [6]. As a result of the study, it was found that the ultimate bearing capacity of the bored piles was estimated by a new SPT direct method. This method gives the highest correlation of all available SPT methods and is validated with soil data. Additionally, there are some methods in which the shear strength parameters of the soil are added to the calculation of the pile's ultimate bearing capacity by passing from the SPT values to the empirical relations [7–13].

Another method used in pile ultimate bearing capacity calculations is the finite element analysis method. This method has been widely used in recent years due to its reliable results [14,15]. In addition, in recent years, studies have been carried out to determine the ultimate bearing capacity of piles using artificial intelligence applications [16,17].

As can be seen from the years of experience in geotechnical engineering, the pile bearing capacity can be determined using the results obtained from the loading tests performed on the piles manufactured at full scale in situ. Of these loading tests, the static loading test is undoubtedly the most reliable method. In determining the ultimate bearing capacity of bored piles, axial compression tests, and axial tensile tests are used to resist buoyancy in piles under groundwater [18]. The axial compression test ASTM D 1143-81 and the tensile test ASTM D 3689-07 are the most widely used standards [19,20].

There are many methods for pile load test evaluation, such as Chin-Kondner, Mazurkiewicz, Decourt, and Ozkan-Alku [21–24]. Loading test studies on 10 piles were carried out by Alku and evaluated with the methods in the literature [24]. Accordingly, the lowest failure load values were obtained by Davisson and De Beer methods, and the highest failure load values were obtained by Chin-Kondner and Decourt methods [21,23,25,26]. As a result of statistical analysis, it was concluded that Davisson and De Beer's methods gave more uncertain results than other methods [25,26]. In the second part of the study, the Ozkan-Alku method was developed [24]. In another study, the results of five different pile loading tests carried out in various parts of the Jakarta region were examined with different methods [27]. The method closest to the results of the pile loading test was found to be the Lastiasih and Sidi methods [28]. The data from 45 pile load tests performed by Mahmood et al. using 8 different methods were statistically interpreted [29]. As a result of the study, it was concluded that the most suitable and reliable pile capacity assessment methods for bored piles are the De Beer and Mazurkiewicz methods [22,26]. In another study, it was concluded that the six-pile loading test results gave 20–40% less value compared to the NAVFAC D.M. 7.02 theoretical method [30,31]. In addition, the limit equilibrium method and the 90% Hansen method gave equal results [32]. It was stated that the best approach is to use the average values of all ultimate pile-bearing capacities.

The bearing capacities of 10 m, 20 m, 30 m, 40 m, and 50 m long piles were investigated by Ghiasi and Eskandari [33] using numerical and theoretical methods. According to the results obtained, it was concluded that the results of the Peck et al. method and the Meyerhof method were similar to the Plaxis 2D results [3,34]. In another study, the results of 22 pile loading tests in three different areas of Nasiriyah were evaluated with analytical methods [35]. Accordingly, Chin-Kondner and Hansen were the methods that gave the highest failure load [21–32]. The Decourt, De Beer, and Mazurkiewicz methods gave the closest mean values to the failure load [5,22,26]. The Butler and Hoy method gave the smallest failure value [36]. In the study conducted by Olgun et al. [37], the ultimate bearing capacity of bored piles manufactured in different soil conditions was determined using load-settlement data using graphical methods [5,21,22,26,32,38,39]. The closest average failure load value was obtained by the Corps of Engineers and the Butler and Hoy methods. In the study conducted by Józefiak et al., the numerical analysis results of CFA piles modeled using Abaqus commercial software were compared with the results of the pile loading test, and it was concluded that the two methods gave consistent results [40]. Studies were carried out using the data from 24 pile loading tests conducted in various regions of Turkey and abroad [41]. As a result of the analysis, the Chin-Kondner method gave the highest ultimate bearing capacity value out of the 12 methods, and the De Beer method gave the

lowest ultimate bearing capacity value [21,26]. In another study conducted by Mert and Ozkan, pile loading test data were evaluated with different methods [42]. In the evaluation, the closest value to the failure value was obtained using the De Beer method [26]. In the study conducted by Dalkilic, 25 different pile loading test results, most of which did not reach the failure load value, were evaluated using 14 different methods [18]. According to the data obtained, the application reliability of the Fuller and Hoy, Hirany and Kulhawy, Hansen, and Davisson methods was found to be very low in the evaluation of pile loading tests that have not reached failure; the lowest bearing capacity value was obtained by the De Beer method, and the highest bearing capacity value was obtained by the Decourt method [5,25,26,32,39,43]. In the study conducted by Henrina et al., according to the data obtained from eight cases where pile loading tests were performed in the Jakarta region, it was concluded that the method closest to the ultimate pile bearing capacity value is the Decourt method [5,44]. In addition, the method developed by Shariatmadari et al. gave the highest estimate [45].

Pile foundation data obtained from the Rusunawa Project were compared with the finite element method, methods based on SPT data, and pile loading tests by Harasid et al. [46]. The closest result to the loading test value (280 tons) was obtained by the Mazurkiewicz method (270 tons) [22]. In recent years, there have been many studies on the determination of pile carrying capacity based on artificial neural networks [47–51]. A feedforward neural network (FFNN) is proposed to determine the ultimate axial load bearing capacity of pre-stressed precast high-strength concrete pile foundations by Nguyen et al. [52]. It has been found that the proposed FFNN method is consistent with the measurement results and is reliable. In addition to artificial neural network methods, methods developed based on statistical results have also been available in recent years [53].

It has been seen that the pile loading test evaluation methods based on nearly 40 mathematical models in the literature give values that are quite far from the failure load determined by theoretical methods and the failure load determined by graphical methods, even in piles up to failure. In this context, a new method development study has been carried out. In this study, a new method has been proposed to determine the ultimate pile bearing capacity by using eight full-scale pile loading tests. The proposed method has been compared with theoretical methods based on SPT data, methods using SPT data indirectly, full-scale pile loading test evaluation methods, and finite element methods.

## 2. Materials and Methods

### 2.1. Study Parameters

In the study, data obtained from 8 pile loading tests conducted in various provinces of Turkey was used (Table 1).

**Table 1.** General information about pile loading tests used in the study.

| Date/Test Pile No | Type of Manufacture | Working Type of Piles | Section Geometry | |
| --- | --- | --- | --- | --- |
| | | | Diameter (D) | Length (m) |
| 1 | Bored | Pressure | 1.20 | 36.00 |
| 2 | Bored | Pressure | 1.00 | 25.00 |
| 3 | Bored | Pull-Out | 0.80 | 18.00 |
| 4 | Bored | Pressure | 1.00 | 20.00 |
| 5 | Bored | Pressure | 1.00 | 34.00 |
| 6 | Bored | Pressure | 0.80 | 26.00 |
| 7 | Bored | Pressure | 0.65 | 25.00 |
| 8 | Driven | Pull-Out | 0.65 | 30.30 |

The ground conditions obtained as a result of the drilling work carried out in the regions where the piles were located are given in Table 2. Since the pile loading test is a test performed in a few days at most, the situation it represents for cohesive soils is the undrained condition. For this reason, undrained parameters were taken into account when

determining the mechanical parameters of fine-grained soils. The undrained shear strength resistance of fine-grained soils was determined by field SPT tests. Since it is impossible to take undisturbed samples in coarse-grained soils, the mechanical parameters of coarse-grained (cohesionless) soils were determined by SPT tests conducted in the soil. In order to determine the mechanical parameters of the rock units, the Hoek-Brown failure criteria were changed to Mohr-Coulomb parameters. The end soil on all piles is clay. Piles 1, 2, 3, 4, and 5 are bored piles in fine-grained soils. Pile 7 is a bored pile that reaches failure in fine-grained soils. Pile 8 is a driven pile in fine-grained soil. Pile 6 is the bored pile in the coarse-grained ground. Test piles 3 and 8 are pull-out piles, and others are pressure piles. All piles have a circular cross-section and are friction/adhesion piles.

**Table 2.** Idealized ground conditions of test piles, which are the basis for finite element calculations and theoretical calculations.

| Test Pile No | Groundwater Level (m) | Ground Conditions | $z$ (m) | $c'$ (kPa) | $c_u$ (kPa) | $\phi'$ (°) | $E_{50}$ (MPa) | $\nu$ | $N_{a(awg)}$ |
|---|---|---|---|---|---|---|---|---|---|
| 1 | ±0.00 | Weathered rock | 0.00–16.00 | 30.53 | - | 19.38 | 387.60 | 0.25 | - |
|  |  | Sandy silty clay | 16.00–36.00 | - | 160 | - | 160 | 0.495 | 38 |
| 2 | ±0.00 | Weathered rock | 0.00–2.50 | 22.22 | - | 22.96 | 405.9 | 0.25 | - |
|  |  | Sandy silty clay | 2.50–7.30 | - | 84 | - | 84 | 0.495 | 20 |
|  |  | Sandy silty clay | 7.30–25.00 | - | 135 | - | 135 | 0.495 | 32 |
| 3 | ±0.00 | Weathered rock | 0.00–2.00 | 22.22 | - | 22.96 | 405.9 | 0.25 | - |
|  |  | Sandy silty clay | 2.50–7.30 | - | 84 | - | 84.0 | 0.495 | 20 |
|  |  | Sandy silty clay | 7.30–18.00 | - | 135 | - | 135.0 | 0.495 | 32 |
| 4 | ±0.00 | Weathered rock | 0.00–8.00 | 18.08 | - | 25.79 | 425 | 0.25 | - |
|  |  | Sandy silty clay | 8.00–11.00 | - | 78 | - | 78 | 0.495 | 18 |
|  |  | Sandy silty clay | 11.00–20.00 | - | 147 | - | 147 | 0.495 | 34 |
| 5 | ±0.00 | Sandy silty clay | 0.00–7.50 | 31.09 |  | 21.25 | 436.4 | 0.25 | - |
|  |  | Sandy silty clay | 7.50–34.00 |  | 185 | - | 185 | 0.495 | 43 |
| 6 | ±0.00 | Weathered rock | 0.00–24.00 | 0 | - | 38 | 50 | 0.30 | - |
|  |  | Sandy silty clay | 24.00–26.00 | - | 180 | - | 180 | 0.495 | - |
| 7 | 6.50 | Clay | 0–2.80 |  | 17 |  | 17 | 0.495 | 4 |
|  |  | Clay | 2.80–8.00 |  | 26 |  | 26 | 0.495 | 6 |
|  |  | Clay | 8.00–11.00 |  | 73 |  | 73 | 0.495 | 17 |
|  |  | Clay | 11.00–19.50 |  | 91 |  | 91 | 0.495 | 21 |
|  |  | Clay | 19.50–22.00 |  | 130 |  | 130 | 0.495 | 30 |
|  |  | Clay | 22.00–25.00 |  | 95 |  | 95 | 0.495 | 22 |
| 8 | 6.00 | Clay | 0–2.95 |  | 13 |  | 13 | 0.495 | 3 |
|  |  | Clay | 2.95–7.65 |  | 26 |  | 26 | 0.495 | 6 |
|  |  | Clay | 7.65–15.15 |  | 69 |  | 69 | 0.495 | 16 |
|  |  | Clay | 15.15–20.15 |  | 104 |  | 104 | 0.495 | 24 |
|  |  | Clay | 20.15–30.30 |  | 117 |  | 117 | 0.495 | 27 |

All piles specified within the scope of the study are reinforced concrete. The elasticity module of the piles and therefore, of the concrete, has been determined in accordance with the Turkish Standard, Code Number TS500, Design and Construction Rules of Reinforced Concrete Structures [54]. According to this standard, the modulus of elasticity of concrete is:

$$E_p = 3250 \cdot \sqrt{f_{ckj}} + 14000 \tag{1}$$

can be calculated with the relation. Here;

　$E_p$: Modulus of elasticity of concrete (pile) (MN/m²),
　$f_{ckj}$: It is the 28-day compressive strength of concrete in MPa (MN/m²).

The Poisson ratio ($\nu_p$) of the concrete (piles) is taken as 0.20 as specified by the TS500 standard.

Some of the parameters assigned for the soil types in the finite element software (Plaxis 2D) are specified in the following items: In finite element software;

The pile-soil interface strength parameter coefficient ($R_{int}$) was taken as 0.70 for soils. 1.00 was taken for weathered rocks.

This value means that the soil strength parameters are reduced at that rate only at the soil-pile interface.

The secant modulus ($E_{50}ref$) in the hardening soil model was accepted as equal to the modulus of elasticity.

The power (m) value was taken as 1.00 in all soil models created with hardening soil.

*2.2. Method*

Within the scope of the study, pile bearing capacity determination methods based on the Standard Penetration Test, methods that use SPT data indirectly, pile loading test evaluation methods based on a mathematical model, and the finite element method, which will be used in the analyses suitable for the soil type and manufacturing method in which the test piles are located, were used.

2.2.1. Direct Standard Penetration Test Methods for Determining Ultimate Pile Bearing Capacity

The Standard Penetration Test, which is widely used in the world, is a test that determines soil density/consistency as well as helping the calculation of pile bearing capacity. The empirical correlation suggestions of some researchers, whose suggestions are widely used in this study, are given below.

The Calculation of pile tip ($q_b$) and surface resistance ($q_u$) is done in the Meyerhof method according to Equation (2). It is a valid method only for coarse-grained soils [3]. In the equation, $N_b$ is the average of the SPT N values between 10D above and 5D below the pile base. It is determined as $n_s = 1$, $k = 0.012$, $m = 0.12$ for bored piles; and as $n_s = 2$, $k = 0.04$, $m = 0.4$ and by using the equations $q_b$ *(MPa)* $= k \cdot N_b$, $q_s$ *(kPa)* $= n_s \cdot N_s$ for driven piles.

$$m \cdot N_b \leq (L/D) \tag{2}$$

The Decourt method calculation formula is given in Equations (3a) and (3b) [5]. For fine-grained soils: $\alpha$ value is 1.0, $k_b$ value is taken as 0.10 for driven piles and 0.08 for bored piles. In coarse-grained soils: $\alpha = 0.5$–0.6 (average value was taken in the study), $k_b = 0.325$, $(N_b)_{60}$: Average of the corrected SPT N values around the pile base, and $(N_s)_{60}$: is the average of the corrected SPT N values along the relevant layer to be calculated.

$$q_b(MPa) = k_b \cdot (N_b)_{60} \tag{3a}$$

$$q_s(kPa) = \alpha \cdot \{2,8 \cdot (N_s)_{60} + 10\} \tag{3b}$$

The correlations valid for the Bazaraa and Kurkur methods for bored piles are expressed in Equations (4a) and (4b) [4]. The $N_b$ value in these equations is the average of the SPT-N values between 1D above and 3.75D below the pile base with the condition of $N_b \leq 50$. $N_s$: It is the average of the SPT N values along the relevant layer to be calculated. $n_b$: 0.06–0.2 (average value was taken in the study); $n_s$: 2–4 (average value was taken in the study).

$$q_b(MPa) = n_b \cdot N_b \tag{4a}$$

$$q_s(kPa) = n_s \cdot N_s \tag{4b}$$

2.2.2. Indirect Standard Penetration Test Methods for Determining Ultimate Pile Bearing Capacity

The methods by which shear strength parameters of the soil are included in the pile's ultimate bearing capacity by passing the SPT values from the SPT values with empirical correlations are called indirect SPT calculations. The undrained shear strength of cohesive soils was used in the soil SPT tests and empirical methods suggested in the literature.

The adhesion factor "$\alpha_u$" for bored piles was proposed by the O'Neil and Reese method as in Equations (5a) and (5b) [9]. $c_u$: undrained shear strength (kN/m$^2$), $p_a$: atmospheric pressure, and $\alpha_u$: adhesion factor.

$$c_u / p_a \leq 1.5 \ ; \ \alpha_u = 0.55 \tag{5a}$$

$$1.5 \leq c_u / p_a \leq 2.5 \ ; \ \alpha_u = 0.55 - 0.1 \cdot (c_u / p_a - 1.5) \tag{5b}$$

The adhesion factor "$\alpha$" for drilled piles was determined by the Kulhawy and Jackson method, and the relation is given in Equation 6 [10]. $c_u$: undrained shear strength (kN/m$^2$), $pa$: atmospheric pressure (101.3 kN/m$^2$), and $\alpha u$: adhesion factor).

$$\alpha = 0.21 + 0.25 \cdot (p_a / c_u) \leq 1.00 \tag{6}$$

The Focht and Vijayvergiya method developed for clayey soils is given in Equation (7) [8]. $\lambda$: Adhesion coefficient, $\sigma'_0$: Average effective stress along the pile (kN/m$^2$), $c_u$: Average undrained shear strength along the pile (kN/m$^2$).

$$f_{s,ult} = \lambda \cdot (\sigma'_0 + 2 \cdot c_u) \tag{7}$$

In coarse-grained (cohesionless) soils, the skin frictional resistance is similar to the lateral earth pressure calculation due to the vertical effective stress and as shown in Equation (8). $f_{s,ult}$: Ultimate surface friction resistance per unit area (kN/m$^2$), $\sigma'_0$: Average vertical effective stress of the calculated layer (kN/m$^2$), $K_s$: Lateral earth pressure coefficient (Unitless), $\delta$: angle of friction between the soil and the pile (°: Degrees); $K_0$: the earth pressure at rest. The ratio of the lateral earth pressure coefficient $K_s$ to the earth pressure coefficient at rest $K_0$ was chosen according to the pile construction method suggested by Tomlinson. Depending on the pile construction method [7].

$$f_{s,ult} = \sigma'_0 \cdot K_s \cdot tan\delta \tag{8}$$

2.2.3. Finite Element Method (Plaxis 2D)

Two-dimensional analysis and axial symmetry were used in the analysis studies. Since there is a sigma(z) effect in axisymmetric analysis, it gives results very close to 3D. The ultimate load capacity, which was examined using finite element software, continued to increase based on the average load increase in the pile loading test in piles that did not reach failure [55]. Since the load-settlement curve is a sharp turning curve in the finite element software, the load failure load is accepted where it starts to plasticize.

2.2.4. Pile Loading Test Evaluation Methods Based on Mathematical Model

The mathematical evaluation of pile loading tests is conducted by making this curve a mathematical model as a result of the operations performed on the load-settlement curve and reaching the failure load by extrapolation over this model.

By generalizing Kondner's studies on stress-strain for all piles, the Chin-Kondner method has been developed to interpret pile loading tests that do not reach failure load by extrapolation [21]. Linearly distributed points are joined on an ideal line. If the point where this line intersects the (settlement/load) axis is called $C_2$, and the incline of the line is called $C_1$, the inverse of the incline of the line gives the pile failure load, as in Equation (9).

$$Q_{ult} = 1/c_1 \tag{9}$$

In the Decourt method, the load value at each stage is divided by its corresponding settlement value, and the obtained values and load values are indicated as a distribution on a graphic [23]. The linear point series of the distribution are combined on an approximate line. If the incline of this approximate line is called $C_1$, the quantity of the point where it intersects the load/settlement axis is called $C_2$, and the failure load is calculated using Equation (10).

$$Q_{ult} = c_2/c_1 \tag{10}$$

The Ozkan-Alku method is to determine the load-settlement curve as an even function, not an odd function [24]. The failure (ultimate) load is found by the end function, not the initial. For the end function, plot the $\sqrt{Q}/\delta - Q$ (square root of load divided by settlement-load). The points in this distribution move linearly after a certain value. The incline of the line formed by combining these points and the point where it intersects the vertical axis are found. The point that intersects the apsethe is the Ozkan-Alkus failure load.

### 2.2.5. Description of The Proposed Method

According to the results of the examination and analysis made in this study, a mathematical model-based method that can evaluate the failure load based on the data of 8 pile loading tests has been proposed. The proposed method has been turned into a mathematical model as a result of the process of the load settlement curve.

The pile loading test evaluation methods based on the mathematical model in the literature give values far away from the failure load determined by theoretical methods and the failure load determined by graphical methods even in piles up to failure. The main reason for this situation is thought to be due to the excessive iteration of the methods.

When the pile loading test curves of the piles which reached failure are examined, it is seen that these curves consist of approximately three separate sections and are shown in Figure 1.

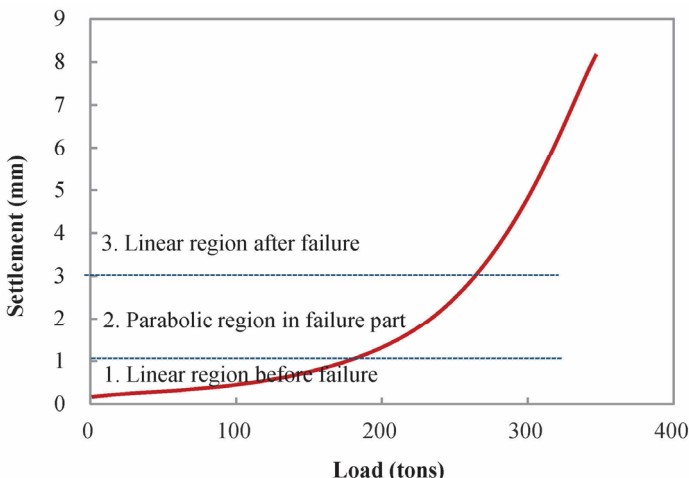

**Figure 1.** Typical pile loading test curves of the piles that reached failure.

By iterating based on the mathematical model, it has been determined that the distribution group to be selected to estimate the pile failure load and the ordinate-apse correlation of the distribution should be determined precisely.

Therefore,

I.　The load-settlement relationship needs to be examined at the logarithmic level.

II.　While extrapolating based on the mathematical model, it has been determined that using the distribution within the linear part before a failure will give a conservative failure load, while using the distribution within the linear section after failure will give an excessive failure load. For this reason, it has been seen that extrapolating over the distribution in the parabola during the load will give the ideal result.

Application of the proposed method:

The application of the proposed method is expressed below:

At each load-settlement intersection, the settlement value $\left(log_{Q_i}^{\Delta_i}\right)$ is found at the algorithm load base.

$\Delta_i$: Settlement value in each loading stage (mm),

$Q_i$: Load value in each loading stage (tons).

The The average of all values $\left(log_{Q_i}^{\Delta_i}\right)$ is calculated: $\frac{\Sigma_{i=1}^{n} log_{Q_i}^{\Delta_i}}{n}$

n: Number of loading stages

For each load stage, $K = \dfrac{Q_i^{\frac{\Sigma_{i=1}^{n} log_{Q_i}^{\Delta_i}}{n}}}{\Delta_i}$ value is calculated (this value is represented by "K" symbol) and K-Q distribution is drawn.

K: It is the correspondence of the value of $\left(\dfrac{Q_i^{\frac{\Sigma_{i=1}^{n} log_{Q_i}^{\Delta_i}}{n}}}{\Delta_i}\right)$ (Unitless).

In the resulting distribution, the distribution formed by the points corresponding to the parabola during the failure, shown in Figure 2, is combined on an ideal line, and the point where this line intersects the apse gives the ultimate bearing capacity (failure load). If it is desired to work more precisely, the regression line is drawn by performing regression analysis, and the equation of the regression line is found as:

$$y = a \cdot x + b \tag{11}$$

a: Incline of the line (tons$^{-1}$),

b: The point where the line intersects the ordinate (y) axis (Unitless)

Failure load is found as:

$$Q_{ult} = -b/a \tag{12}$$

where:

a: Incline of the line(tons$^{-1}$),

b: The point where the line intersects the ordinate (y) axis (Unitless).

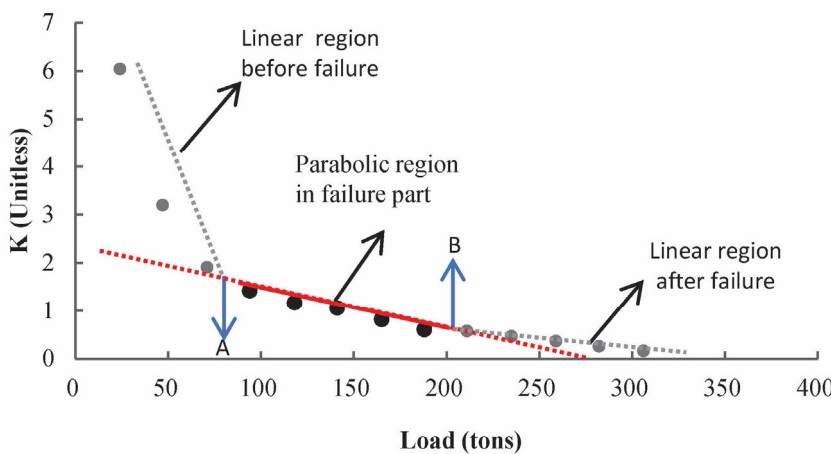

**Figure 2.** Typical graph of the proposed method for piles up to failure.

As can be seen in Figure 2, the distribution of the pile loading test of the piles that reached failure has two breaking points, A and B, and three different trendlines. Since there is no linear part after failure in piles that do not reach failure, the distribution formed by the parabola part during failure will be clearly seen with this method. Thanks to the related equation, the "linear region before failure", "linear region after failure," and "linear region after failure" intersection corners can be seen more clearly in the load-settlement graph.

Sample application for the proposed method:

The ultimate bearing capacity of the pile loading test, which is given below, is determined by using the proposed method described in detail. In Figure 3, the load-settlement graph of the sample application is given.

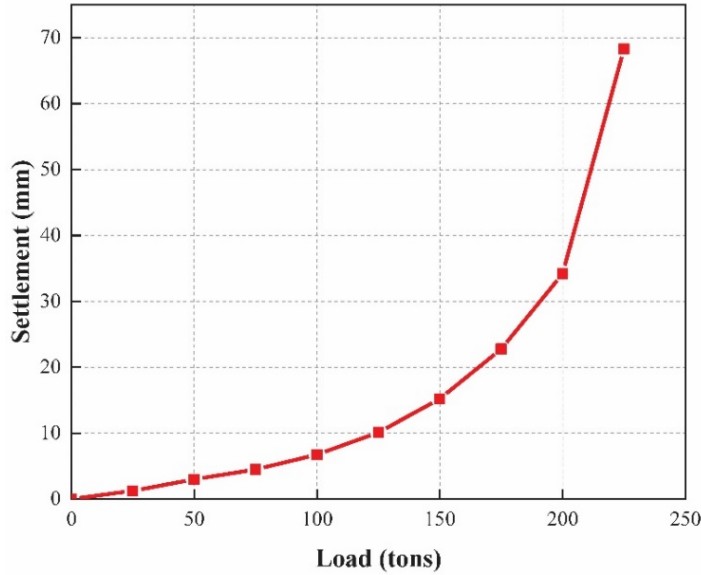

**Figure 3.** Load-settlement graph of the sample application.

The distribution of $Q_i \sim K$ created for all stages according to the proposed method is given in Table 3. The distribution graph required for the solution is divided into three parts in the sample application in Figure 4, as indicated in the theory of the proposed method. In the distribution, the breaks in the 2nd and 6th loading stages were clearly seen, and the interpretation was made accordingly.

**Table 3.** The $Q_i \sim K$ distribution generated for all steps according to the proposed method.

| Process Step | Settlement $\Delta$(mm) | Load Q (tons) | $\left(log_{Q_i}^{\Delta_i}\right)$ | $\frac{\sum_{i=1}^{n}log_{Q_i}^{\Delta_i}}{n}$ | $K=\frac{Q_i^{\frac{\sum_{i=1}^{n}log_{Q_i}^{\Delta_i}}{n}}}{\Delta_i}$ |
|---|---|---|---|---|---|
| - | 0.000 | 0.00 | - | | - |
| 1 | 1.250 | 25.00 | 0.0693 | | 3.5768 |
| 2 | 3.000 | **50.00** | **0.2808** | | **2.0575** |
| 3 | 4.500 | **75.00** | **0.3484** | | **1.6564** |
| 4 | 6.750 | **100.00** | **0.4147** | | **1.2625** |
| 5 | 10.125 | **125.00** | **0.4795** | **0.4653** | **0.9337** |
| 6 | 15.188 | 150.00 | 0.5429 | | 0.6776 |
| 7 | 22.781 | 175.00 | 0.6052 | | 0.4853 |
| 8 | 34.172 | 200.00 | 0.6665 | | 0.3443 |
| 9 | 68.344 | 225.00 | 0.7800 | | 0.1818 |
| | | | **Total: 4.1873** | | |
| | | | **Number of samples (n): 9** | | |

1. Stage: The expression of settlement at the logarithm load base is calculated (Equation (13)).

$$log_{100}^{6,75} = 0.4147 \tag{13}$$

2. Stage: The settlement expressions on the logarithm load base are summed (Equation (14)).

$$\frac{\sum_{i=1}^{n} log_{Q_i}^{\Delta_i}}{n} = \frac{0.0693 + 0.2808 + 0.3484 + 0.4147 + 0.4795 + 0.5429 + 0.6052 + 0.6665 + 0.78}{9} = 0.4653 \tag{14}$$

$$K = \frac{Q_i^{\frac{\sum_{i=1}^{n} log\frac{\Delta_i}{Q_i}}{n}}}{\Delta_i} = \frac{100^{0.4653}}{6.750} = 1.2625 \tag{15}$$

3.    Stage: The coefficient defined by the symbol K is found (Equation (15)).

A value of 1.265 was found for a load of 100 tons. In all steps, these processes are applied in order, and the $Q_i \sim K$ scat graph is formed for all stages (Figure 4).

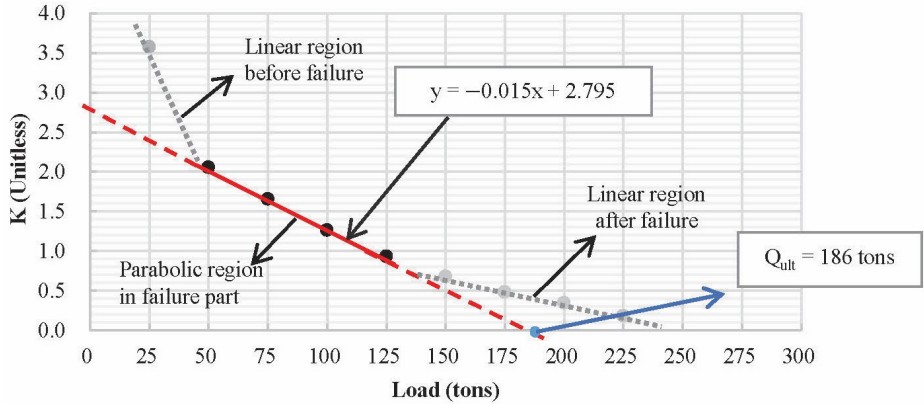

**Figure 4.** The solution graph of the sample application for the proposed method.

The distribution graph required for the solution is divided into three parts in the sample application, as stated in the theory of the proposed method. In the distribution, the breaks in the second and sixth loading stages are clearly visible. The intersection points from the first to the second break were interpreted as the parabola during the failure, and the ideal line was drawn by regression analysis on these points. The ultimate bearing capacity (failure load) at the point where this line intersects the apse was found to be 186 tons. It can also be found by using Equation (16) as follows:

$$Q_{ult} = -b/a = -2.795/-0.015 = 186 \text{ tons} \tag{16}$$

## 3. Results and Discussion

### 3.1. Finding the Ultimate Pile Bearing Capacity Directly by Standard Penetration Test

The application of the Meyerhof method to test pile 6 is shown in Figure 5 [3].

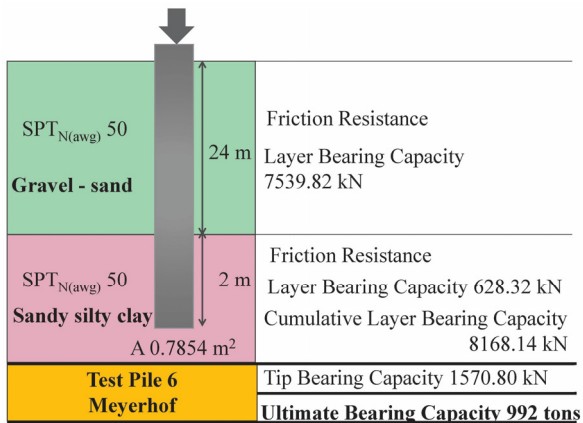

**Figure 5.** Ultimate bearing capacity calculation of test pile 6 according to the Meyerhof method [3].

The application of the Decourt method and Bazaraa and the Kurkur method on test pile 1 is shown in Figure 6 [4,5].

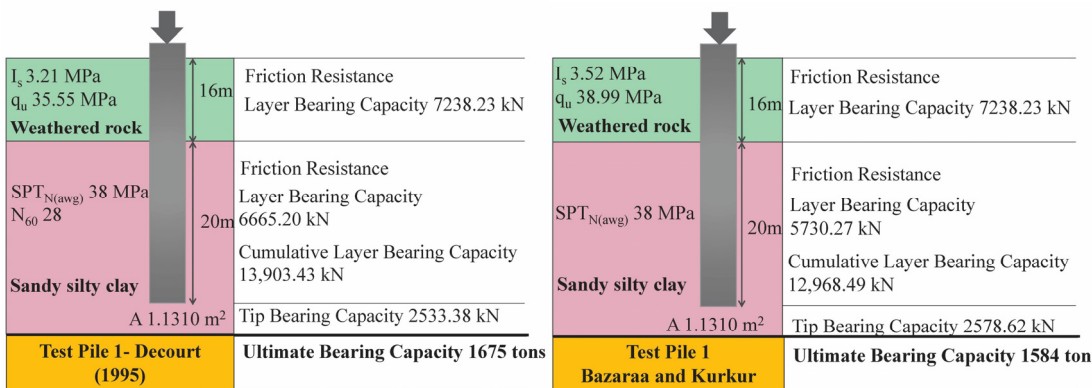

**Figure 6.** Ultimate bearing capacity calculation of test pile 1 according to the Decourt method and the Bazaraa and Kurkur method [4,5].

### 3.2. Finding the Ultimate Bearing Capacity of the Pile by Indirect Standard Penetration Test

The application of the O'Neil and Reese method and the Kulhawy and Jackson method on test pile 1 is shown in Figure 7 [9,10].

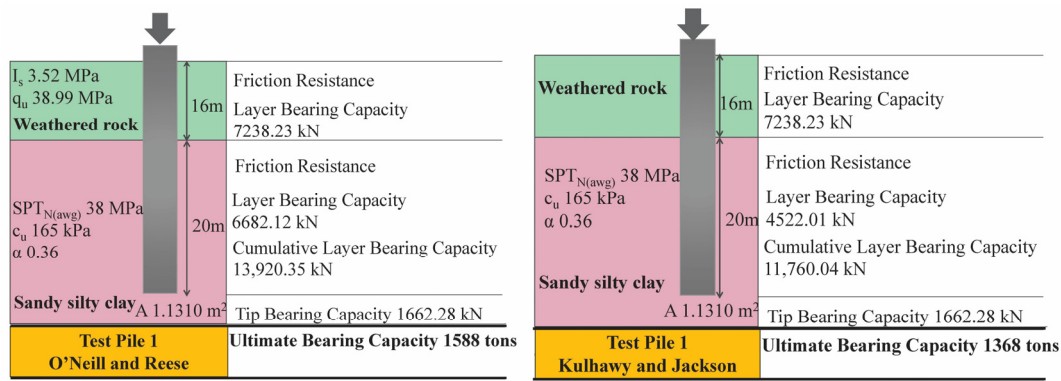

**Figure 7.** Ultimate bearing capacity calculation of test pile 1 according to O'Neil and Reese method and Kulhawy and Jackson method [9,10].

The application of the Vijayvergiya and Fotch method and the Tomlinson method on test pile 1 is shown in Figure 8 [7,8].

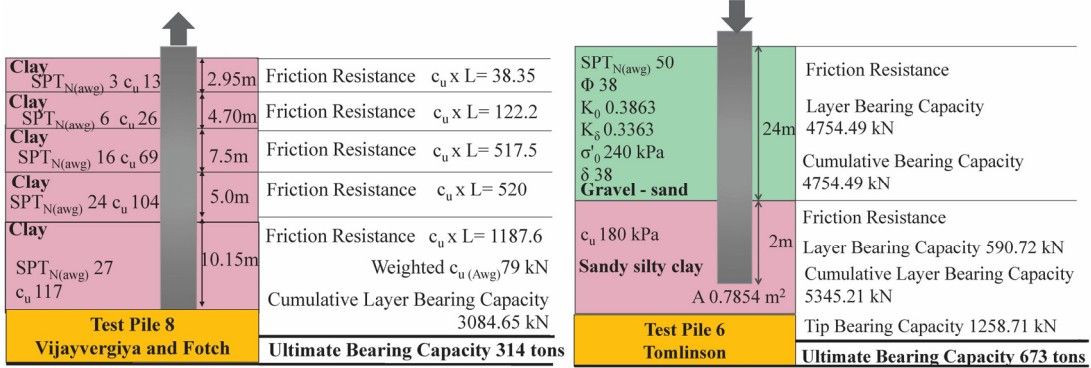

**Figure 8.** Ultimate bearing capacity calculation of test pile 8 according to Vijayvergiya and Fotch method and test pile 6 according to Tomlinson method [7,8].

### 3.3. Finite Element Method (Plaxis 2D)

The sample regarding the application of the method was determined to be 1350 tons for test pile 1, and the load-settlement curve is shown in Figure 9.

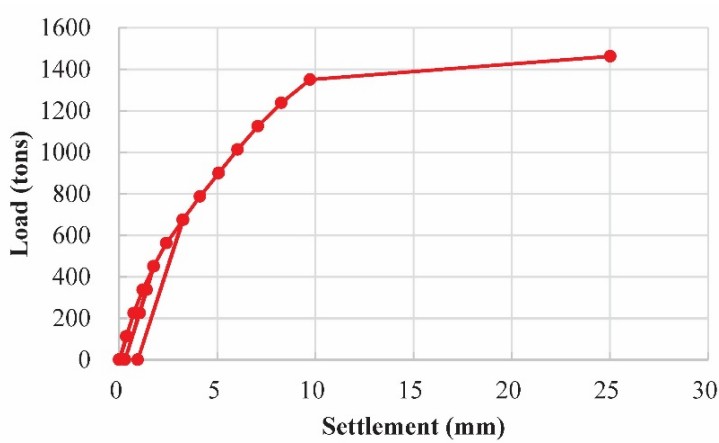

**Figure 9.** Finding test pile 1 by the finite element method.

*3.4. Evaluation Methods of Pile Loading Test Based on Mathematical Models*

The load-settlement curves obtained from the test pile loading test are given in Figure 10. Some of the piles were loaded up to the failure load, and some of them could not pass from the linear load-settlement curve to the parabola load-settlement curve.

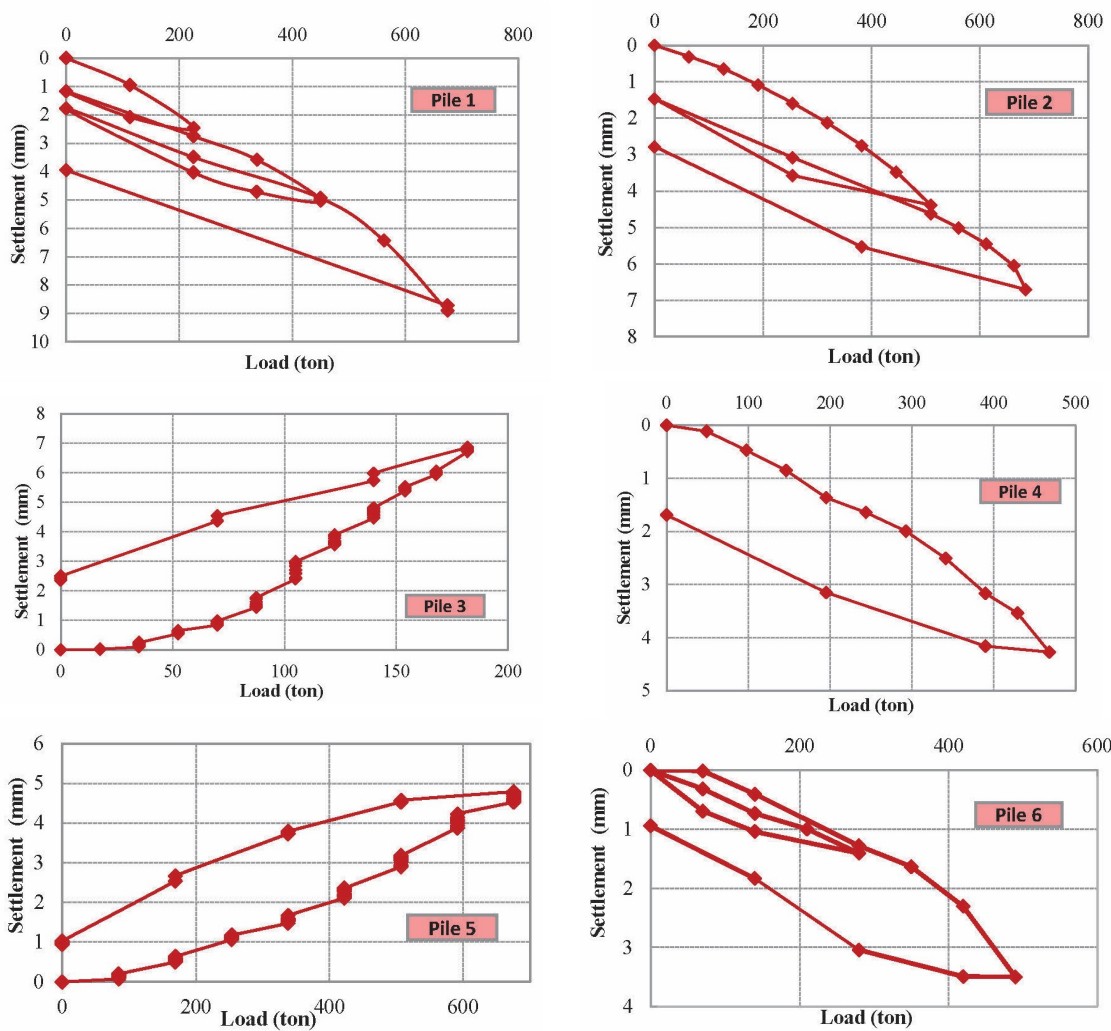

**Figure 10.** *Cont*.

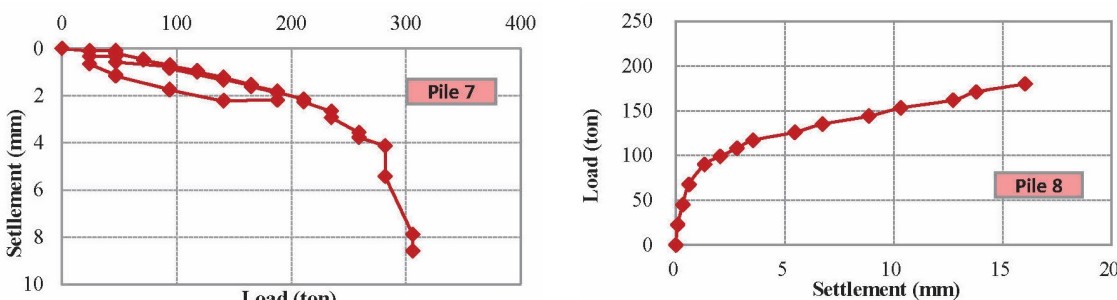

**Figure 10.** Load-settlement graphs of test piles.

Calculation examples based on pile loading test data for test pile 1 are given in Figure 11.

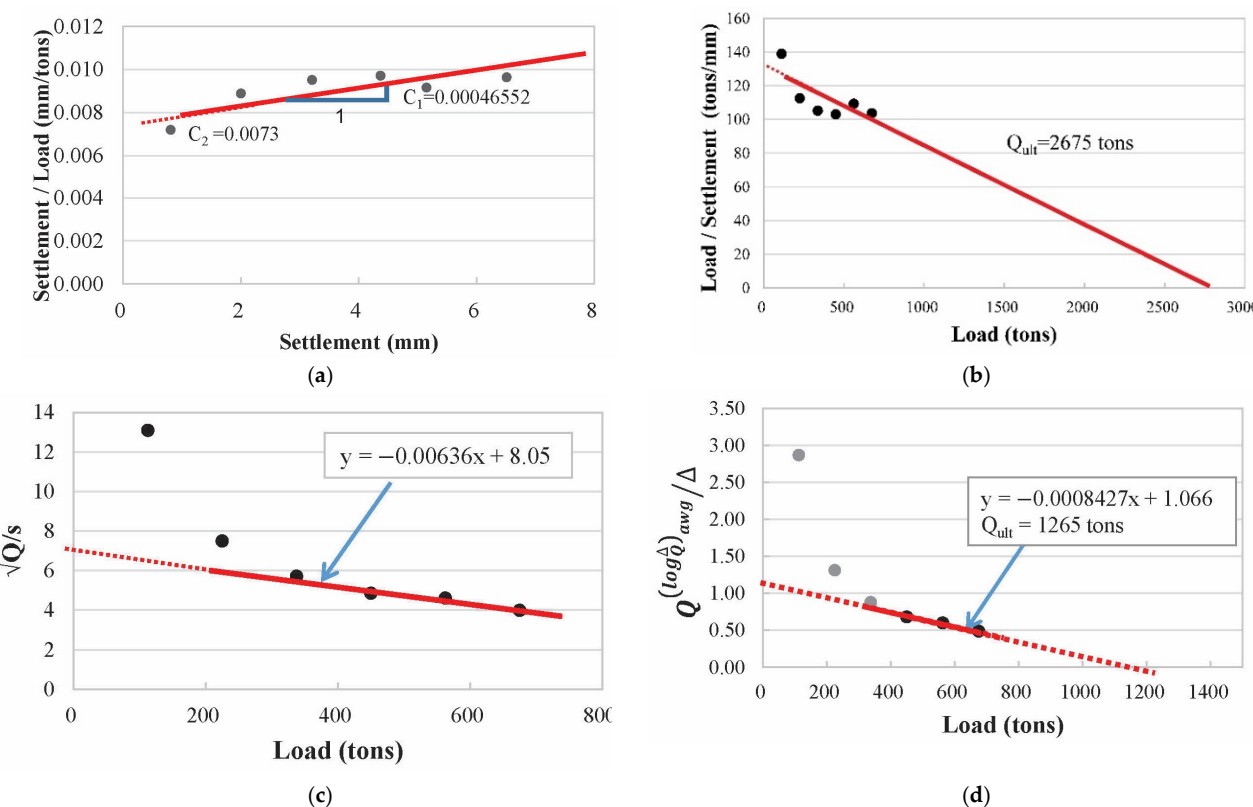

**Figure 11.** Pile loading test calculations based on a mathematical method on test pile 1. (**a**) Chin [21]; (**b**) Decourt [23]; (**c**) Alku [24]; (**d**) Recommended Method.

The Ultimate bearing capacity values obtained as a result of the calculations made according to the methods on 8 test piles are given in Figure 12. According to the evaluations made for test piles 1, 2, 3, 4, 6, 7 and 8, when all methods were evaluated together, the closest value to the mean value was found as the Ozkan-Alku method for test pile 1, Decourt method for test piles 2, 3 and 8., Alku method for test pile 5, Bazaraa and Kurkur and O'Neil and Reese methods for test pile 5, Decourt (1995) and Decourt (1999) methods for test pile 6, and the Plaxis 2D method for test pile 7 [4,5,9,23,24,55].

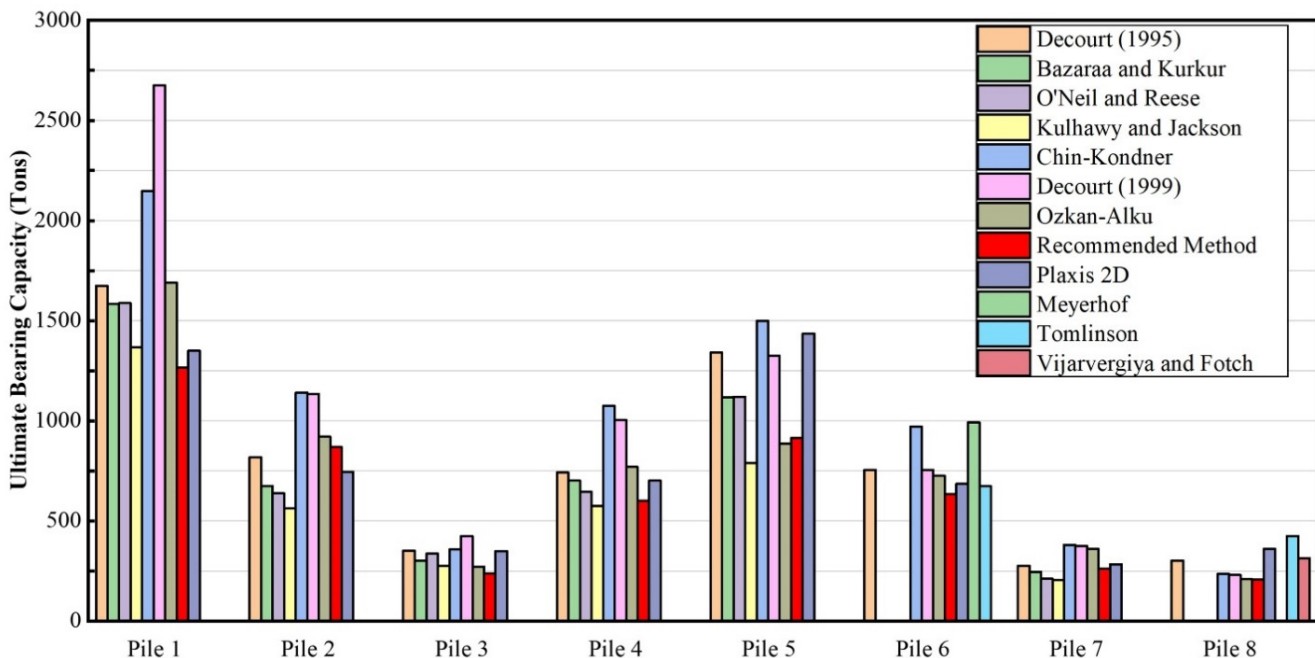

**Figure 12.** Ultimate bearing capacity calculation results for 8 test piles according to the methods [3–5,7–10,21,23,24,55].

The ratios of each method to the average ultimate bearing capacity of each pile are given in Figure 13. When all methods were evaluated together, the closest failure load value to the proposed method was determined by the Kulhawy and Jackson method for test pile 1, the Ozkan-Alku method for test piles 2, 3, 5, and 8, the Tomlinson method for test pile 6, and the Decourt method for test pile 7 [5,7,10,24].

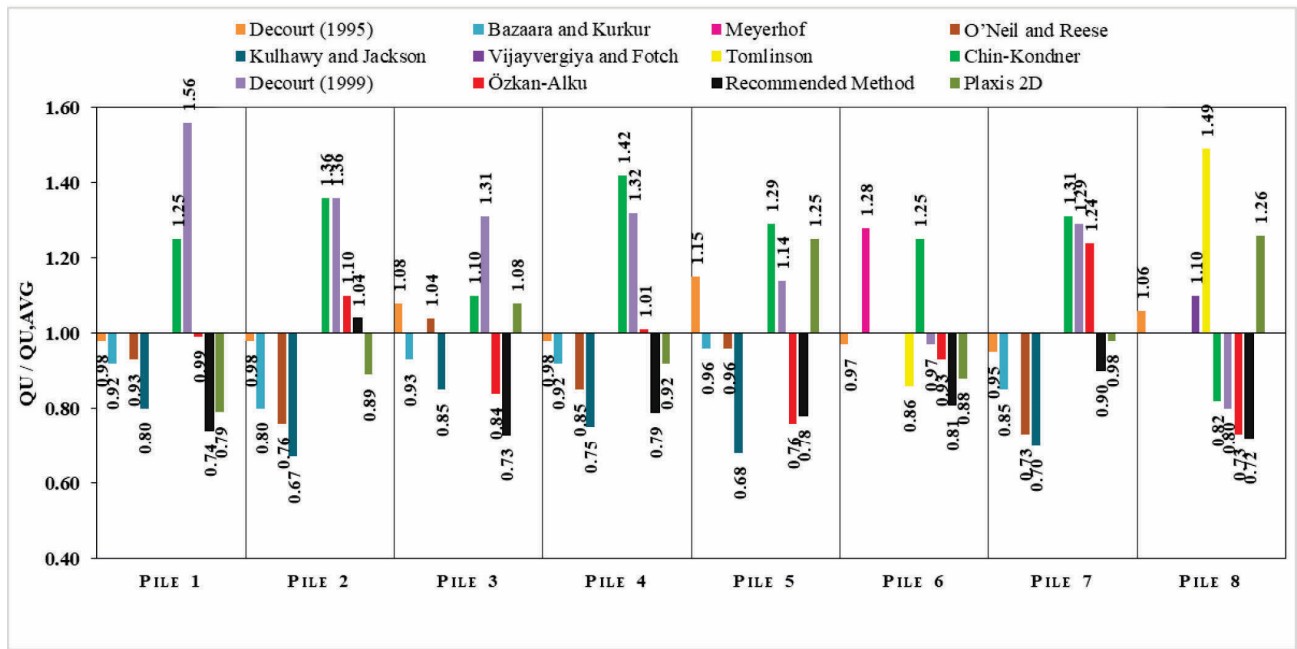

**Figure 13.** Comparison of the ultimate bearing capacity values with the average values according to each test pile method [3–5,7–10,21,23,24,55].

Among the theoretical methods, the closest results to the proposed method were found in the Kulhawy and Jackson method for test piles 1, 3, 4, 5, the Decourt method for test piles 2 and 8, the Tomlinson method for test pile 6, the Decourt method, and the Bazaraa and Kurkur method for test pile 7 [4,5,7,10].

According to pile loading test evaluation methods, the Ozkan-Alku method gave the closest failure load value to the proposed method for all test piles [24].

Compared to the finite element method, the proposed method gave 1.94 times higher failure load values for test pile 1, 1.17 times higher values for test pile 2, 0.68 times higher values for test pile 3, 0.85 times higher values for test pile 4, 0.64 times higher values for test pile 5, 0.93 times higher values for test pile 6, 0.92 times higher values for test pile 7, and 0.57 times higher values for test pile 8.

When all methods were evaluated together for bored piles in fine-grained soils (test piles 1, 2, 3, 4, and 5), the highest failure load was obtained in the evaluation method based on the mathematical model of Decourt [23]. The Kulhawy and Jackson methods gave 45.6% lower results [10]. The proposed method, on the other hand, yielded 40.2% lower results. The Chin-Kondner method gave the highest failure load when all methods were evaluated for the individual test pile 7, which collapsed in fine-grained soils [21]. The proposed method was 31.3% lower, and the Kulhawy and Jackson method gave the lowest result with 46.3% [10]. The Tomlinson method gave the highest failure load in the results obtained for the individual test pile 8, which was manufactured as driven in fine-grained soils [7]. The proposed method yielded the lowest failure load, with a rate of 51.5%. The Meyerhof method gave the highest failure load in the results obtained for individual test pile 6, which was manufactured as bored piles in coarse-grained soils [3]. The lowest value was obtained by the proposed method with a rate of 36.1%.

As a result of the evaluations, although the proposed method gave results close to the Alku method among the pile loading test evaluation methods, since the load-settlement values of the linear section after failure were not used in piles that collapsed, excessive iteration was prevented [24]. Accordingly, the proposed method, as the graphical method, offers a more reasonable value of failure load, which is close to the truth in the parabola part during failure.

The descriptive statistical values of the ultimate bearing capacity values calculated for the 8 methods and the proposed method in the study were calculated for each method and shown in Table 4. When Table 4 is examined, it is seen that the highest average ultimate bearing capacity values were calculated by the Decourt method with 990.6 tons and the Chin-Kondner method with 976 tons, respectively [21,23]. It is seen that the lowest average ultimate bearing capacity values were calculated by the proposed method with 623.4 tons and the Kulhawy and Jackson methods with 628.8 tons, respectively [10].

**Table 4.** Descriptive statistical values for the bearing capacity calculation methods and the proposed method.

| Methods | Number of Data | Value Range (tons) | Minimum Value (tons) | Maximum Value (tons) | Avarage (tons) | Standard Error (tons) | Standard Deviation (tons) | Variance (ton²) |
|---|---|---|---|---|---|---|---|---|
| Chin-Kondner [21] | 8 | 1913.0 | 235.0 | 2148.0 | 976.0 | 230.3 | 651.5 | 424,442.0 |
| Decourt (1999) [23] | 8 | 2445.0 | 230.0 | 2675.0 | 990.6 | 277.2 | 784.0 | 614,710.3 |
| Ozkan-Alku [24] | 8 | 1482.0 | 209.0 | 1691.0 | 729.0 | 169.2 | 478.6 | 229,052.6 |
| Recommended Method | 8 | 1059.0 | 206.0 | 1265.0 | 623.4 | 134.4 | 380.1 | 144,465.7 |
| Decourt (1995) [5] | 8 | 1398.0 | 277.0 | 1675.0 | 783.0 | 178.5 | 504.8 | 254,861.7 |
| Bazaraa and Kurkur [4] | 6 | 1338.0 | 246.0 | 1584.0 | 770.7 | 207.5 | 508.4 | 258,442.3 |
| O'Neil and Reese [9] | 6 | 1377.0 | 211.0 | 1588.0 | 764.2 | 209.3 | 512.7 | 262,867.0 |
| Kulhawy and Jackson [10] | 6 | 1164.0 | 204.0 | 1368.0 | 628.8 | 171.7 | 420.6 | 176,893.8 |
| Plaxis 2D [55] | 8 | 1152.0 | 284.0 | 1436.0 | 739.0 | 156.2 | 441.7 | 195,062.6 |

When the correlation table (Table 5) was examined, it was seen that there were very high and positive correlations between the methods. The highest correlation coefficient (0.998) was between Bazaraa and Kurkur and O'Neil and Reese methods at a high level, while the lowest correlation coefficient was between Ozkan-Alku and finite element method (Plaxis 2D) methods with 0.837, which indicated a high level of coefficient [4,9,24,55]. On the other hand, it was understood that the highest correlation coefficient between the proposed approach and the other approaches was between the Chin-Kondner method with 0.984 and the finite element method (Plaxis 2D) with 0.911 [21,55].

**Table 5.** Correlation coefficients between the examined bearing capacity methods and the proposed bearing capacity method.

|  | Chin-Kondner | Decourt (1999) | Ozkan-Alku | Recommended Method | Decourt (1995) | Bazaraa and Kurkur | O'Neil and Resee | Kulhawy and Jackson | Plaxis 2D |
|---|---|---|---|---|---|---|---|---|---|
| Chin-Kondner [21] | 1 | 0.966 | 0.977 | 0.984 | 0.983 | 0.989 | 0.983 | 0.979 | 0.929 |
| Decourt (1999) [23] | 0.966 | 1 | 0.983 | 0.944 | 0.946 | 0.968 | 0.967 | 0.994 | 0.851 |
| Ozkan-Alku [24] | 0.977 | 0.983 | 1 | 0.97 | 0.936 | 0.949 | 0.943 | 0.974 | 0.837 |
| Recommended Method | 0.984 | 0.944 | 0.97 | 1 | 0.964 | 0.951 | 0.95 | 0.941 | 0.911 |
| Decourt (1995) [5] | 0.983 | 0.946 | 0.936 | 0.964 | 1 | 0.993 | 0.994 | 0.966 | 0.972 |
| Bazaraa and Kurkur [4] | 0.989 | 0.968 | 0.949 | 0.951 | 0.993 | 1 | 0.998 | 0.987 | 0.938 |
| O'Neil and Reese [9] | 0.983 | 0.967 | 0.943 | 0.95 | 0.994 | 0.998 | 1 | 0.986 | 0.939 |
| Kulhawy and Jackson [10] | 0.979 | 0.994 | 0.974 | 0.941 | 0.966 | 0.987 | 0.986 | 1 | 0.874 |
| Plaxis 2D [55] | 0.929 | 0.851 | 0.837 | 0.911 | 0.972 | 0.938 | 0.939 | 0.874 | 1 |

An ANOVA test was applied to the ultimate bearing capacity averages, and there are two hypotheses regarding this test:

$H_0$: There is no difference among group means ($p > 0.05$);

$H_1$: There are significant differences among group means ($p \leq 0.05$).

It was investigated whether there was a difference between the ultimate bearing capacity values obtained for the eight methods calculated according to the data obtained as a result of the studies and the bearing capacity values obtained for the proposed method by performing multiple comparison tests at the 95% confidence interval (Table 6). Accordingly, it was seen that there was no difference in the 95% confidence interval between the proposed method and the other methods, and the significance ($p$) value for all methods was much greater than 0.05.

**Table 6.** Variance analysis detail for bearing capacity values calculated with the proposed method and other methods.

| (I) Method | (J) Methods | Avarage Difference (I−J) | Standard Error | Significance ($p$) | 95% Confidence Interval | |
|---|---|---|---|---|---|---|
|  |  |  |  |  | Lower Limit | Upper Limit |
| Recommended Method | Chin−Kondner [21] | −352.63 | 269.25 | 0.924 | −1220.65 | 515.40 |
|  | Decourt (1999) [23] | −367.25 | 269.25 | 0.906 | −1235.27 | 500.77 |
|  | Ozkan−Alku [24] | −105.63 | 269.25 | 1.000 | −973.65 | 762.40 |
|  | Decourt (1995) [5] | −159.63 | 269.25 | 1.000 | −1027.65 | 708.40 |
|  | Bazaraa and Kurkur [4] | −147.29 | 290.82 | 1.000 | −1084.86 | 790.28 |
|  | O'Neil and Reese [9] | −140.79 | 290.82 | 1.000 | −1078.36 | 796.78 |
|  | Kulhawy and Jackson [10] | −5.46 | 290.82 | 1.000 | −943.03 | 932.11 |
|  | Plaxis 2D [55] | −115.63 | 269.25 | 1.000 | −983.65 | 752.40 |

In the study conducted by Adel and Shakir, different methods were compared in the results obtained from the pile loading test studies conducted in different regions of the city of Nasiriyah [35]. The proposed method was also compared with the results of Hansen, Decourt, Chin-Kondner, Mazurkiewicz, Fuller-Hoy, Buttler-Hoy, and Tangant

methods. The recommended method in the TP1 loading test was 1.13 times the average with 879 tons; in the TP2 test with 1172 tons, 0.95 times the average; in the TP7 test with 551 tons, 1.19 times the average; in the TP8 test with 585 tons, it was 1.10 times the average; and in the TP9 loading test, 1.05 times the average result was obtained with 526 tons.

Each pile loading test evaluation method has its own philosophy. But all of them serve a single purpose, and that purpose is to interpret the load-settlement curve as close to reality as possible. Therefore, it is usual to make comparisons between these methods. The results of a new pile loading test conducted in Turkey according to both the proposed method and Chin-Kondner were evaluated and compared (Table 7).

**Table 7.** Idealized load-settlement data.

| Settlement $\Delta$ (mm) | Load Q (tons) |
| --- | --- |
| 0.00 | 0 |
| 0.98 | 10 |
| 1.39 | 30 |
| 2.08 | 80 |
| 2.82 | 150 |
| 3.49 | 250 |
| 4.71 | 450 |

According to the proposed method, the ultimate pile bearing capacity was determined to be 897.35 tons (Figure 14).

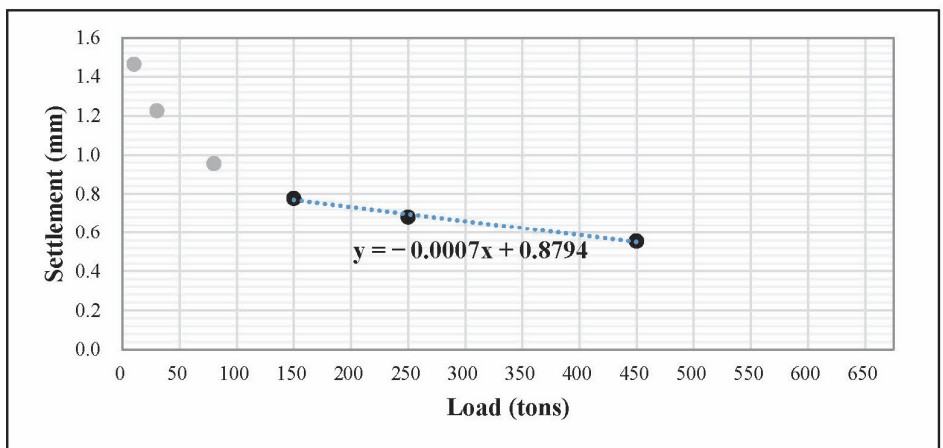

**Figure 14.** Evaluation according to the proposed method.

However, in the Chin-Kondner method, which is the sample method taken from the literature for comparison, even a curve suitable for the criteria of the method was not formed (Figure 15).

There may be two reasons for this situation: either the Chin-Kondner (1963) method is insufficient or the pile needs to be loaded more in order to determine the ultimate bearing capacity according to the Chin-Kondner method. With comparison, the proposed method is superior in terms of both reliability and economy, and the targets are achieved.

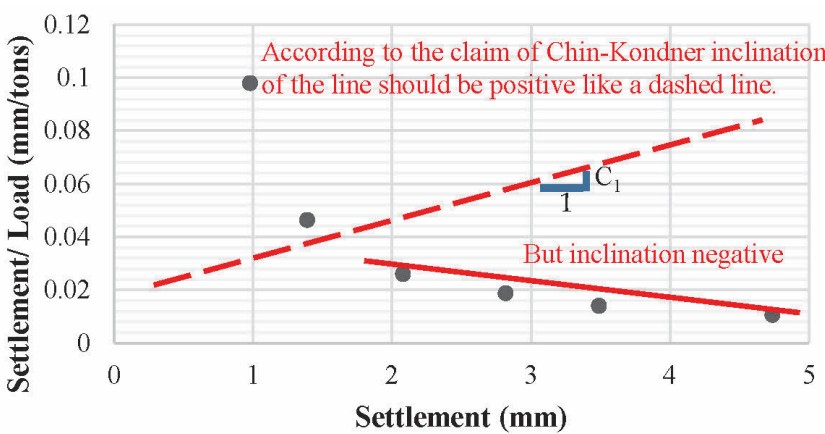

**Figure 15.** Evaluation according to the Chin-Kondner method.

## 4. Conclusions

In the study, the data obtained from eight different pile loading experiments was examined. Pile loading tests were analyzed for ultimate bearing capacity by using the theoretical methods (based on SPT data, indirectly based on SPT data), evaluation methods of pile loading tests, and the finite element method based on soil and laboratory tests performed on the ground where the test pile was located. The dependence of the proposed method as a distribution was examined, and it was determined that the strongest linear correlation with a coefficient of 0.984 was obtained by the Chin-Kondner method [21]. This situation only expresses the linear dependence, and the methods that gave the closest results together with the proposed method in terms of the averages of the failure load values (quantities) were the Kulhawy and Jackson method with a 9.0% difference in bored piles in fine-grained soils, the Bazaraa and Kurkur method with a 6% difference in bored piles that reached failure in fine-grained soils (Test Pile 7), and the Ozkan-Alku method with a difference of 1.5% (Test Pile 8) in driven piles in fine-grained soil and Tomlinson method with a 6.2% difference in the bored pile in coarse-grained soil (Test Pile 6) [4,7,10,24]. Multiple comparison tests of the proposed method with other methods were performed, and it was determined that it remained within the 95% confidence interval according to the applied variance analysis. Based on this situation, it was concluded that the proposed method can be used in both coarse-grained and fine-grained soils.

As a result of the comparisons made, although the proposed method gives results close to the Ozkan-Alku method, excessive iteration is prevented since the load-settlement values of the linear section after failure are not used in piles that reach failure. Accordingly, the proposed method offers a more reasonable value of failure load, which is close to the truth in the parabola in the failure region.

Other important advantages of the proposed method are that it shortens the test time and is economical, as the results will be obtained earlier than with other pile-bearing test evaluation methods.

**Author Contributions:** Conceptualization, İ.V. and H.K.; methodology, H.K.; validation, İ.V., H.K. and S.P.; formal analysis, H.K.; investigation, H.K.; resources, S.P. and H.K.; data curation, H.K.; writing—original draft preparation, H.K. and S.P.; writing—review and editing, S.P.; visualization, H.K. and S.P.; supervision, İ.V. All authors have read and agreed to the published version of the manuscript.

**Funding:** This research received no external funding.

**Institutional Review Board Statement:** Not applicable.

**Informed Consent Statement:** Not applicable.

**Data Availability Statement:** Data are contained within the article. The data presented in this study can be requested from the authors.

**Conflicts of Interest:** The authors declare no conflict of interest.

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
