# Peer review of "A Novel Approach Proposal for Estimation of Ultimate Pile Bearing Capacity Based on Pile Loading Test Data"

_applsci, doi:10.3390/app13137993_

Round 1

Reviewer 1 Report

The following should be added for further consideration:

1-    There are missing studies on the topic; these studies compared the results of load tests and ML techniques with the standard correlations. Hence, please include the following studies

Evolutionary computing to determine the skin friction capacity of piles embedded in clay and evaluation of the available analytical methods. Transportation Geotechnics, 24, 100372. DOI: 

Development of an optimized model to compute the undrained shaft friction adhesion factor of bored piles. Geomechanics and Engineering, 28(4), pp.397-404. DOI:

A new development of ANFIS–GMDH optimized by PSO to predict pile bearing capacity based on experimental datasets. Eng Comput 2019:1–16.

Modelling load–settlement behaviour of piles using high-order neural network (HON-PILE model). Eng Appl Artif Intell 2011;24(5):813–21.

Vertical bearing capacity of the pile foundation with restriction plate via centrifuge modelling. Ocean Eng 2019;181:109–20.

New method for predicting the ultimate bearing capacity of driven piles by using Flap number. KSCE J Civ Eng 2015;19(3):611–20.

Optimizing an ANN model with ICA for estimating bearing capacity of driven pile in cohesionless soil. Eng Comput 2018;34(2):347–56.

2-    Table 2: I suggest you change soil properties column to ground conditions as weathered rock is not soil and it is not correct to be listed under a column named soil properties.

3-    You need to explicitly state that your method is solely based on the load settlement curve.

4-    Please add a section to explain how the shear strength parameters in Table 2 has been calculated in the previous studies. This is important these parameters might vary based on the equipment used (direct shear or triaxial) and this will affect the accuracy of the analytical correlations.

5-    Important, your method is based on an assumption based on pile load test (load-settlement) and thus, it might not provide an advanced in the topic as each of the available methods has its own philosophy to interpret the load settlement behavior and extract the ultimate load from it. This should be discussed in the paper to shed light on this important aspect. 

Author Response

Dear Rewiever 1;
The study has been arranged in line with the suggestions of you esteemed rewievers and other rewievers. The answers to your comments are given below. The current file of the study is attached.

Best regards

Response to Reviewer 1 Comments

Point 1: 1-    There are missing studies on the topic; these studies compared the results of load tests and ML techniques with the standard correlations. Hence, please include the following studies

  1. Evolutionary computing to determine the skin friction capacity of piles embedded in clay and evaluation of the available analytical methods. Transportation Geotechnics, 24, 100372. DOI: 
  2. Development of an optimized model to compute the undrained shaft friction adhesion factor of bored piles. Geomechanics and Engineering, 28(4), pp.397-404. DOI:
  3. A new development of ANFIS–GMDH optimized by PSO to predict pile bearing capacity based on experimental datasets. Eng Comput 2019:1–16.
  4. Modelling load–settlement behaviour of piles using high-order neural network (HON-PILE model). Eng Appl Artif Intell 2011;24(5):813–21.
  5. Vertical bearing capacity of the pile foundation with restriction plate via centrifuge modelling. Ocean Eng 2019;181:109–20.
  6. New method for predicting the ultimate bearing capacity of driven piles by using Flap number. KSCE J Civ Eng 2015;19(3):611–20.
  7. Optimizing an ANN model with ICA for estimating bearing capacity of driven pile in cohesionless soil. Eng Comput 2018;34(2):347–56.

Response 1: It has been added to the introduction part of the suggested study examples (except e) on the subject (page 3, paragraph 3).

Point 2:  Table 2: I suggest you change soil properties column to ground conditions as weathered rock is not soil and it is not correct to be listed under a column named soil properties.

Response 2: Soil properties expression in Table 2 has been changed to ground condition. It is shown by marking in Table 2.

Point 3:  You need to explicitly state that your method is solely based on the load settlement curve.

Response 3: In the summary of the study, it is emphasized that the proposed method is based on the load-settlement curve.It is shown by marking in abstract. In addition, the relevant emphasis has been made in the first paragraph under the title number 2.2.5. It is shown by marking in the first paragraph under the title number 2.2.5.

 Point 4:  Please add a section to explain how the shear strength parameters in Table 2 has been calculated in the previous studies. This is important these parameters might vary based on the equipment used (direct shear or triaxial) and this will affect the accuracy of the analytical correlations.

Response 4: The shear parameters of cohesive/non-cohesive soils and rock units are explained in paragraph 2 under title 2.1. It is shown by marking in the second paragraph under the title number 2.2.5.

Point 5: Important, your method is based on an assumption based on pile load test (load-settlement) and thus, it might not provide an advanced in the topic as each of the available methods has its own philosophy to interpret the load settlement behavior and extract the ultimate load from it. This should be discussed in the paper to shed light on this important aspect. 

Response 5: Of course, each pile loading test evaluation method has its own philosophy. But all of them serve a single purpose and that purpose; is to interpret the load-settlement curve as close to reality as possible. Therefore, it is natural to make comparisons between these methods. The results of a new pile loading test conducted in Turkey according to both the proposed method and Chin-Kondner (1963) were evaluated and compared. (Included in the results and discussion section).

Reviewer 2 Report

This paper presents a new method for estimating the ultimate bearing capacity of pile foundations based on eight pile load tests carried out in Türkiye's provinces. This method was compared with other methods, like theoretical methods, finite element analysis methods, and pile load test evaluation methods. Through analysis of variance, the conclusion is drawn that the new method maintains a 95% confidence interval. Overall, it is a relatively new method for determining bearing capacity.

The following modifications must be made before acceptance:

1) The new method is based on bearing capacity testing, while numerical simulation and SPT in other comparative methods do not require pile foundation bearing capacity testing. Will the cost of the new method too high?

2) Since the bearing capacity test has been conducted, further clarification is needed on the advantages of the new method compared to the code method. It is recommended to add the local pile foundation industry code in Türkiye.

3) What is the meaning of logiQ, how is it derived, and does it have a specific physical meaning?

4) What is the meaning of Figure 13? Does the author want to use a new method that is closer to the average to demonstrate the rationality of the new method. So assuming that other methods have significant errors, their average values will also have significant errors. If the new method approaches this average value, does it mean that the new method has significant errors?

5The author uses eight piles in Türkiye to verify the rationality of the method in this paper, which is only used as a verification of rationality. But no more piles were estimated, which does not meet the "estimation" in the title. Please add literature and compare the ultimate bearing capacity of pile foundations with other literature to further verify the accuracy of the method proposed in this paper.

Author Response

Dear Rewiever 2;
The study has been arranged in line with the suggestions of you esteemed rewievers and other rewievers. The answers to your comments are given below. The current file of the study is attached.

Best regards

Response to Reviewer 2 Comments

Point 1: The new method is based on bearing capacity testing, while numerical simulation and SPT in other comparative methods do not require pile foundation bearing capacity testing. Will the cost of the new method too high?

Response 1: Cost comparison can only be made according to pile test evaluation methods. Compared to other pile bearing strength test evaluation methods, the test period will be shorter and more economical since results will be obtained earlier. These advantages have been added to the last paragraph under the conclusion title and shown by marking.

Point 2: Since the bearing capacity test has been conducted, further clarification is needed on the advantages of the new method compared to the code method. It is recommended to add the local pile foundation industry code in Türkiye.  

 Response 2: There is no local code used in Turkey. According to ASTM, it is recommended to use methods that can be used in the literature. Also, code, norms and standards are not the main subject of our research. Our aim is to make pile loading tests more economical and reliable.

Point 3: What is the meaning of logiQ, how is it derived, and does it have a specific physical meaning? 

Response 3: Thanks to the related formula, we can more clearly see the junction corner parts of "linear region before failure", "parabolic region in failure part" and "linear region after failure" in the load-settlement graph. Based on this, we were able to make a more reliable extrapolation. It has been added under the title number 2.2.5 Application of the Proposed Method by marking it in the last paragraph.

Point 4: What is the meaning of Figure 13? Does the author want to use a new method that is closer to the average to demonstrate the rationality of the new method. So assuming that other methods have significant errors, their average values will also have significant errors. If the new method approaches this average value, does it mean that the new method has significant errors?

 Response 4: In Figure 13, the distance-closeness relationship of the proposed method and other methods according to the mean reference value is examined. The distance from the mean of some other methods is also shown. The fact that the proposed method is close to the mean shows that it is reliable.

Point 5: The author uses eight piles in Türkiye to verify the rationality of the method in this paper, which is only used as a verification of rationality. But no more piles were estimated, which does not meet the "estimation" in the title. Please add literature and compare the ultimate bearing capacity of pile foundations with other literature to further verify the accuracy of the method proposed in this paper.

Response 5: These are the conclusions we reach on the data we can use. Although the number of studies is relatively small, we think that the proposed method will give results close to reality in induction. At this stage, it is not possible for us to test experiments with more load-settlement data (due to permits required, time constraints, etc.).

In line with your suggestion, the results of a recent pile loading test conducted in Turkey were evaluated with the proposed method and the Chin-Kondner method, and a comparison was made and the inadequacy of the chin method was revealed. This situation is shown by adding the discussion to title “Result and Discussion” section.

Reviewer 3 Report

My comments to the submitted paper:

·       Fig.1 - correct the text - not through the curve

·       Tab.6 should not be divided into 2 pages

·       Tab. 1 -mPa or MPa

·       missing reference [49]

·       Standardize the writing of the decimal point

·       Fig.10 --- unify the load and settlement axes in all graphs

Author Response

Dear Rewiever 3;
The study has been arranged in line with the suggestions of you esteemed rewievers and other rewievers. The answers to your comments are given below. The current file of the study is attached. Best regards.

Response to Reviewer 3 Comments

Point 1: Fig.1 - correct the text - not through the curve.

 Response 1: The expression "during failure" in Figure 1 has been corrected and Figure 1. has been changed. Also, the expressions in figure 2 and figure 4 have been corrected and Figure 2. and Figure 4. has been changed.

Point 2:  Tab.6 should not be divided into 2 pages

Response 2: Table 6 is arranged on one page.

Point 3:  Tab. 1 -mPa or MPa

Response 3: The suggested correction has been made and is shown by marking in Table 1.

Point 4:  missing reference [49]

Response 4: The suggested correction has been made and is marked in the references.

Point 5: Standardize the writing of the decimal point

Response 5: The suggested correction has been made and decimal expressions in figures, tables and formulas are separated by commas in all texts. Relevant corrections (in texts, figures, tables and formulas) are indicated by marking.

Point 6:   Fig.10 --- unify the load and settlement axes in all graphs

Response 6: In all graphs in figure 10, the load settlement axes are unified. It is shown in Figure 10.

Reviewer 4 Report

This very interesting paper concerns a very challenging geotechnical problem, determination of pile bearing capacity. The authors propose a useful high-level test evaluation method. The test has been validated against a number of tests conducted in Turkey.

Perhaps the primary mathematical challenge arises from proper modelling of the soil properties and the interaction of the pile (here, reinforced concrete). My only concern is that the simulation physics are hidden under the Turkish Standard and the finite element code used (Plaxis 2D, by Bentley). This limits the usefulness of this paper somewhat since the reader has to reach for external sources. I'd like the authors to add a section outlining the standard constitutive models used here.

Two additional minor comments:

1. Please comment on statistical modelling of soil properties. In my opinion it is clear that recent advances in stochastic finite element methods will eventually make their way to challenging problems first and then to standard practice.

2. Current software and modelling is understandably limited to 2D. Comment on what 3D would bring in terms of reliability and how it would affect the proposed model.

 No comment.

Author Response

Dear Rewiever 4;
The study has been arranged in line with the suggestions of you esteemed rewievers and other rewievers. The answers to your comments are given below. The current file of the study is attached.

Best regards

Response to Reviewer 4 Comments

Point 1: Perhaps the primary mathematical challenge arises from proper modelling of the soil properties and the interaction of the pile (here, reinforced concrete). My only concern is that the simulation physics are hidden under the Turkish Standard and the finite element code used (Plaxis 2D, by Bentley). This limits the usefulness of this paper somewhat since the reader has to reach for external sources. I'd like the authors to add a section outlining the standard constitutive models used here.

Response 1: Finite element (Plaxis 2D) and TS500 reinforced concrete pile standard descriptions added by marking them below first paragraph on page 5.

Point 2:  Please comment on statistical modelling of soil properties. In my opinion it is clear that recent advances in stochastic finite element methods will eventually make their way to challenging problems first and then to standard practice.

Response 2: In the last paragraph on page 17, an evaluation has been made according to the ground conditions in which the piles are located.

Point 3: Current software and modelling is understandably limited to 2D. Comment on what 3D would bring in terms of reliability and how it would affect the proposed model.

Response 3: Analysis 2D but axisymmetric model. Since there is a sigma(z) effect in axisymetric analysis, it gives results very close to 3D. It is also stated in the plaxis reference manual that there is a 3D-like model. It is indicated by marking in the first sentence in the Finite Element Method (Plaxis 2D).

Round 2

Reviewer 2 Report

1) As a universal method, the prediction method in this article requires more engineering cases besides verifying one's own case. Adding only one comparison is not enough. More pile foundations need to be compared to other projects.

2) Is it more suitable for sandy soil or soft soil? Moreover, it is necessary to clearly indicate which literature has been referenced for pile foundation engineering.

Author Response

Dear Rewiever 2;
The study has been arranged in line with the suggestions of you esteemed rewievers. The answers to your comments are given below. The current file of the study is attached.

Best regards

Point 1: As a universal method, the prediction method in this article requires more engineering cases besides verifying one's own case. Adding only one comparison is not enough. More pile foundations need to be compared to other projects.

Response 1: Your suggestion has been added. It is marked on page 19.

Point 2: Is it more suitable for sandy soil or soft soil? Moreover, it is necessary to clearly indicate which literature has been referenced for pile foundation engineering.

Response 2: It has been added by marking in the Conclusions section.
